# Physical Modelling of Blue Mussel Dropper Lines for the Development of Surrogates and Hydrodynamic Coefficients

**Jannis Landmann** [1,*], **Thorsten Ongsiek** [1], **Nils Goseberg** [2], **Kevin Heasman** [3], **Bela H. Buck** [4,5], **Jens-André Paffenholz** [6] and **Arndt Hildebrandt** [1]

1. Ludwig-Franzius-Institute for Hydraulic, Estuarine and Coastal Engineering, Leibniz Universität Hannover, 30167 Hannover, Germany; ongsiek@lufi.uni-hannover.de (T.O.); hildebrandt@lufi.uni-hannover.de (A.H.)
2. Department of Hydromechanics and Coastal Engineering, Leichtweiß-Institute for Hydraulic Engineering and Water Resources, Technische Universität Braunschweig, 38106 Braunschweig, Germany; n.goseberg@tu-braunschweig.de
3. Cawthron Institute, 7010 Nelson, New Zealand; kevin.heasman@cawthron.org.nz
4. Alfred Wegener Institute for Polar and Marine Research, Faculty of Biosciences, Shelf Sea Systems Ecology, Marine Aquaculture, 27570 Bremerhaven, Germany; bela.h.buck@awi.de
5. Faculty 1, Applied Marine Biology and Aquaculture, Bremerhaven University of Applied Sciences, 27568 Bremerhaven, Germany
6. Geodetic Institute, Leibniz Universität Hannover, 30167 Hannover, Germany; paffenholz@gih.uni-hannover.de
* Correspondence: landmann@lufi.uni-hannover.de; Tel.: +49-511-762-2580

**Abstract:** In this work, laboratory tests with live bivalves as well as the conceptual design of additively manufactured surrogate models are presented. The overall task of this work is to develop a surrogate best fitting to the live mussels tested in accordance to the identified surface descriptor, i.e., the Abbott–Firestone Curve, and to the hydrodynamic behaviour by means of drag and inertia coefficients. To date, very few investigations have focused on loads from currents as well as waves. Therefore, tests with a towing carriage were carried out in a wave flume. A custom-made rack using mounting clamps was built to facilitate carriage-run tests with minimal delays. Blue mussels (*Mytilus edulis*) extracted from a site in Germany, which were kept in aerated seawater to ensure their survival for the test duration, were used. A set of preliminary results showed drag and inertia coefficients $C_D$ and $C_M$ ranging from 1.16–3.03 and 0.25 to 1.25. To derive geometrical models of the mussel dropper lines, 3-D point clouds were prepared by means of 3-D laser scanning to obtain a realistic surface model. Centered on the 3-D point cloud, a suitable descriptor for the mass distribution over the surface was identified and three 3-D printed surrogates of the blue mussel were developed for further testing. These were evaluated regarding their fit to the original 3-D point cloud of the live blue mussels via the chosen surface descriptor.

**Keywords:** aquaculture; drag; inertia; Abbott–Firestone Curve; laboratory tests

---

## 1. Introduction

In recent decades, aquaculture production has served as an essential source of protein for large parts of the world population. An ever-increasing demand for aquatic products suggests that aquaculture will continue to be one of the fastest-growing sectors for the production of protein-based foods [1]. An important part of its production is of bivalve origin, i.e., clams, oysters, mussels and other species. As of 2014, more than 16 million tons of bivalve produce were farmed around the

world [1]. Bivalve cultivation continues to play an important role in providing food for the growing world population.

Currently, mussel farming is situated near or inshore, mostly because sheltered sites have less technological requirements. However, these sites can also pose a variety of different problems. These range from navigational issues for marine vessels [2], nutrient depletion in the wake of mussel farms and undesirable changes in the species assemblage to negative alterations to the benthic environment below the farms due to the mussels' pseudofaeces and marine litter [3]. These issues in combination with more and more contested space nearshore have stipulated a push towards offshore developments. The perspective to use larger volumes of high-quality water to decrease the stress on cultured organisms offshore and the chance to create more revenue by offsetting seafood deficits have led to an increasing effort to move mussel farming into open waters [4].

Aquaculture design is predominantly governed by environmental constraints, and depending on the in situ conditions, different aspects have to be considered in, near or offshore. Especially at offshore sites, high energy is acting on the structures as a result of wave and current conditions. Although hydrodynamic forces acting on fixed and floating structures in such conditions are commonly investigated, existing research related to shellfish aquaculture is generally limited [5]. The feasibility of moving bivalve-related aquaculture offshore is mainly evaluated in regard to farming techniques that are already in use. Suspended farm systems can be divided into intertidal, raft and long-line cultures. Intertidal farms use lines connected to stakes driven into the intertidal sea bed from which the dropper lines are suspended [6]. In raft systems, the mussels are grown on ropes which are hung from a moored raft into the water column [7,8]. Furthermore, so-called "long-line systems" are a possible candidate to be moved further offshore [9]. Long-line systems consist of floatation elements that are connected by ropes to form a mussel-bearing backbone. The backbone is kept in place at each end via an anchor warp connected to the mooring systems. The mussels are cultivated on ropes, so called "dropper lines" or "collectors", and suspended perpendicular from the backbone. The dropper lines are usually spaced evenly along the backbone, with lengths between 5–30 m depending on the water depth and available nutrition. The hydrodynamic forcing on long-line systems has been observed in near-shore conditions to identify the dominant modes of flow-structure interaction and to provide a baseline for designs of future structures [10]. On this basis, work regarding the physics of offshore bivalve-aquaculture has been presented [11]. A description of the hydrodynamic implications of large long-line farms based on observations and scaling arguments is available [12]. A study using a rigid, artificial mussel crop rope constructed from the shells of *Perna canaliculus* provides $C_D$-values for the towing velocities of 0.05 to 0.40 m/s.

The aim of the study was to determine the influence of the inhalants and exhalents of fluids on the drag [13]. An investigation regarding the drag coefficients of suspended canopies, i.e., mussel dropper lines, derived from a physical model with circular cylinders as a representation of the dropper lines is available by Plew, D.R. [14]. Research regarding numerical simulations of suspended bivalve farms is available to a limited extent and mostly details the flow patterns in the wake of farm systems [15]. A numerical study by Raman-Nair and Colbourne [16] to quantify the loads induced into long-line systems is the basis for a numerical model of the three-dimensional dynamics of a submerged mussel long-line system presented by the same author [17]. Further, a flow analysis around mussels is part of the current research activities, however mainly focusing on naturally bedded mussel cultures [18] or on netted structures in association with biofouling [19] in contrast to the long-line systems. While observations of long-line systems are available alongside numerical work, to date no laboratory tests are available. However, experimental data are of crucial concern for the calibration of numerical models facilitating the design process. Thus, this work depicts the first detailed examination of the response of a live-mussel long-line to hydrodynamic forcing in laboratory conditions.

The available literature shows that research gaps concerning the behaviour of suspended long-lines in current and wave conditions exists. Especially the behaviour of dropper lines under waves has received only little attention. A series of physical tests with full-scale blue mussel dropper

lines was conducted to evaluate the corresponding drag and inertia characteristics in the future as well as to determine the magnitude of wave and current forces acting on the mussel dropper lines. To that end, a custom-made test rack consisting of aluminium profiles was attached to a traversing carriage located over a wave flume. Three different specimens of live blue mussel (*Mytilus edulis*) dropper lines were fitted to the test rack. These were tested at various current speeds simulated by the carriage runs to obtain the drag coefficients. Independently, wave tests with varying parameters were conducted to obtain the inertia coefficients. Additionally, surrogate models were created from a digital model based on the 3-D scanned data of the live-mussel dropper lines. The live and surrogate shellfish dropper lines were compared subsequently. The surrogate models were evaluated regarding the statistical mean values of sampled single mussels as well as a surface descriptor. By this means, a number of different surrogates were created and assessed in regard to their fit to the corresponding Abbot–Firestone curve.

## 2. Materials and Methods

Experiments involving live-mussel specimens were carried out at the medium wave and towing tank "Schneiderberg" (WKS) at the Ludwig-Franzius Institute for Hydraulic, Estuarine and Coastal Engineering of the Leibniz Universität Hannover, Germany. The WKS is 110 m long, is 2.2 m wide and contains up to 1.1 m water with a total depth of 2.0 m. In this facility, regular and irregular waves can be generated up to 0.5 m in wave height. A towing carriage allows for testing at constant velocities of up to 1.5 m/s. A plan and side view of the WKS including the towing carriage and wave maker are outlined in Figure 1. All data chosen for the current velocities and wave characteristics were related to potential offshore sites, e.g., off the coast of New Zealand or Canada, and scaled down to allow for future experiments with scaled surrogates. A 1:10 scale was selected, and a Froude similarity was applied.

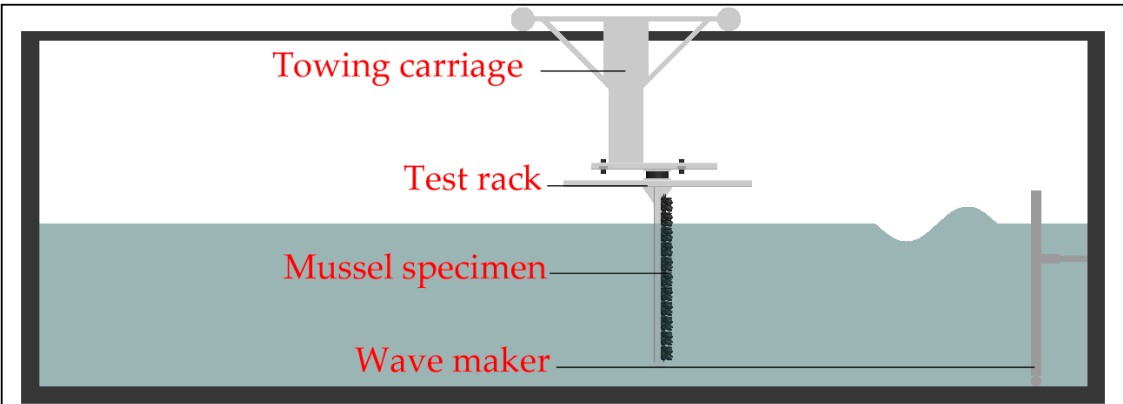

**Figure 1.** A side view of the "Schneiderberg" (WKS) wave tank with a towing carriage, test rack, attached mussel specimen and wave maker (right end); the sketch is not to scale.

A vertical aluminium beam of 2.52 m length, attached to the carriage, allowed the connection of additional equipment. In this work, a load frame was fixed to the main beam that consists of interconnected aluminium profiles. To ensure a sufficient rigidity, a 1.08 m × 0.72 m aluminium frame, horizontally oriented, was used as the top part of the rack. This frame was braced and reinforced by two 1.0 m-long aluminium profiles on the inside of the frame, running in line with the 1.08 m-long outer profiles. The lower part of the test rack was also assembled with an aluminium frame of 1.00 m × 0.80 m, vertically oriented, and used primarily for the attachment of the live-mussel specimens. Figure 2 presents schematic sketches of the test rack with the attached measuring equipment. The main beam, not visible in the sketches, is represented by a connecting plate. Custom-made clamps with a coarse interlocking grid were CNC-milled. These allowed for an easy connection of the mussel dropper line to the test rack while also being able to restrict unwanted movements for the test duration. The clamps were attached to the bottom of the lower frame and the

interior profiles of the upper frame. This allowed for a horizontal orientation of the mussel dropper lines in a defined distance from the main beam. Four wire connections ran from the outer corners of the upper frame to the bottom of the lower frame to further increase the stability. Further, the wires were tensioned to ensure the stability of the test rack. The test rack, as a whole, provided a structure for mussel droppers with a maximal length of 1.1 m. The carriage speed for drag testing was checked via an incremental rotary encoder (SICK DBV50) with a resolution of 12.5 pulses/mm. The rotary encoder allowed for the exact determination of the associated towing velocity by means of single differentiation over the travelled distance. Testing was conducted in either direction, along-flume; hence the rotary encoder installed on the carriage track was either used as a trailing or guiding encoder, depending on the direction of the test run.

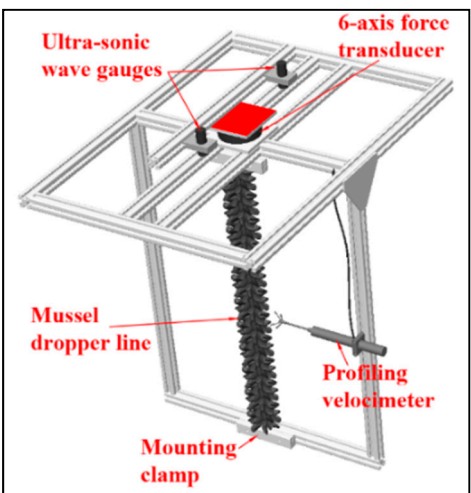

**Figure 2.** A conceptual sketch of the test rack with the measuring equipment and mussel specimen attached in free view: The red plate represents the connection to main beam.

Most importantly, the forces on the mussel dropper lines were of interest when subjected to waves and currents. To obtain this information on the test rack, a six-axis force-transducer (K6D110 ME-Meßsysteme GmbH, 16761 Henningsdorf, Germany) was rigidly attached to the main beam. The K6D110 has a nominal force range of up to 10 kN in x-, y- and z-directions and a nominal torque range of 750 Nm in Mx, My and Mz. Accurate measurements as low as 0.1 kN were possible, based on the specifications by the manufacturer. The x-direction corresponds to a movement from the point of origin along the flume; the y-direction represents the lateral direction, while the z-direction described the direction towards the bottom of the flume. The recorded momentum forces in Mx, My and Mz describe the torques around these axes, respectively. Furthermore, a profiling, acoustic Doppler velocimeter, ADV, (Vectrino Profiler, Nortek, 1351 Rud, Norway) with a maximum sample rate of 100 Hz was attached to the test rack. With the ability to profile three-component velocities over a vertical range of 3 cm and with a resolution of 1 mm, insights into the turbulence dynamics alongside the mussel dropper lines becomes possible. The ADV Profiler was attached in a distance of approximately 0.45 m from the lowest aluminium profile. That approximately corresponds to the midsection of the tested blue mussel dropper lines. Furthermore, measurements of the time-history of the free surface elevation in close vicinity to the dropper line were carried out by ultrasonic wave gauges to obtain information concerning the local wave field. To this end, two locations roughly 0.20 m in front of and behind the dropper line are selected.

All tests were recorded via two sets of cameras. A GoPro Hero4 with a high-definition resolution and a sample rate of 100 fps as well as a Logitech C920 webcam with high-definition resolution and a sample rate of 30 fps were added to the test setup. A linear Field-of-View (FOV) setting for the GoPro Hero4 (Firmware v5.00) was used to correct the convex distortion of the camera. The C920 recorded with a linear FOV as the default setting. The GoPro was installed underwater to the side of the test rack

with a slight offset to the dropper line. The offset ensured an unobstructed view on the test specimen. The webcam was installed on the upper frame of the test rack directed at the blue mussel dropper line in the direction of the wave flume. It was fitted to monitor the wake and oscillatory behaviour of the dropper line. During inertia testing, the carriage, as a whole, was positioned in front of a window section of the flume. A tripod holding another Logitech C920 webcam was used. All video files recorded allowed hindsight into the testing conditions and also provided data regarding the motion response of the dropper-lines under different current velocities. The water depth was kept constant at 0.93 m and was recorded together with temperature periodically throughout the testing procedure. The constant water depth allowed for continuous and identical testing conditions. Figure 3 shows the test rack with the attached measuring equipment and mussel dropper line.

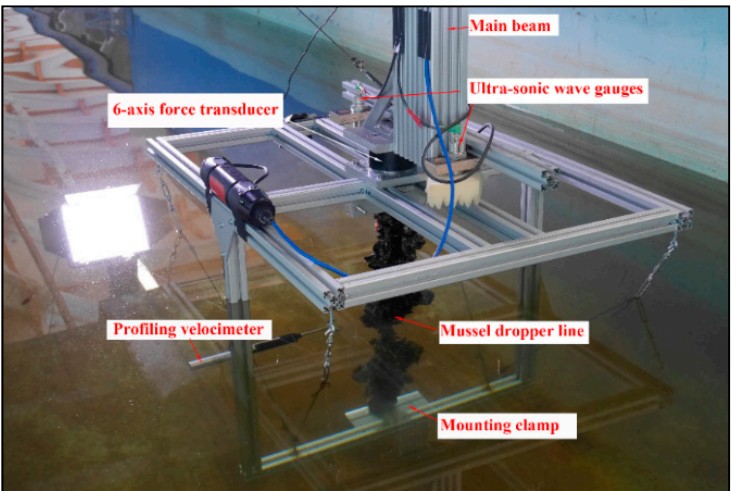

**Figure 3.** The test rack with the Vectrino II Profiler, ultrasonic wave gauges and visible mounting clamps with the attached blue mussel specimen viewed from inside the flume.

A 3.62 m long dropper line of marketable, adult blue mussels (Mytilus edulis) encrusted with newly seeded spat was selected for testing. The blue mussel was selected due to its wide distributional pattern. Furthermore, blue mussels are prime candidates for aquaculture [20]. This is mainly due to its ability to withstand wide fluctuations in salinity, desiccation, temperature and oxygen levels [21]. Blue mussels are found throughout European waters, occupying habitats from Russia to the Bay of Biscay off the French coast. Its zonatial range stretches from the high intertidal to subtidal regions, and its salinity adaptability extends from estuarine areas to fully oceanic seawaters. The blue mussel is euryhaline and proliferates in offshore sea- as well as in brackish-waters down to 4% salinity. Blue mussels are also eurythermal, withstanding freezing conditions for several months. The species is acclimated for a 5–20 °C temperature range, with an upper sustained thermal tolerance limit of about 29 °C for fully grown conditions [21]. These behavioural characteristics allow for a broad application of the insights gathered into drag and inertia characteristics. The blue mussels used in the drag and inertia tests were obtained from an aquaculture farm located in Kiel, Germany and were transported and stored in a controlled sea water tank with aeration and temperature regulation systems in use. Thus, the survival of the mussels for a prolonged time was possible with no loss of adhesive qualities of the byssus, the bundled filaments secreted by the bivalves. The byssus function was the attachment points to the dropper line and was weakened when subjected to adverse conditions.

The blue mussel dropper line was segmented into three specimens with lengths of 1.03 m to 1.05 m. These were labelled 1 to 3 to distinguish in between testing. The specimens were individually weighed, and their width was measured in steps of 5 cm. This was required to determine the mean diameter of the blue mussel specimens. The volumetric displacement of each specimen was determined by immersing them into a container of known dimensions. The water level was then measured before and after the immersion of the specimen. The extant blue mussel dropper line with a length below

50 cm was treated the same way. Additionally, the length, diameter, displacement and weight of individual blue mussels were also determined. Some mussel individuals were selected randomly to obtain information on the individuals of the encrusted rope.

The mean values for the individual mussels correspond to a mean single mussel length *msml* = 4.7 cm, a mean single mussel thickness *msmt* = 2.2 cm and a mean single mussel weight *msmw* = 9.1 g. Figure 4 depicts a specimen prepared before testing alongside a single mussel evaluated regarding the mean values.

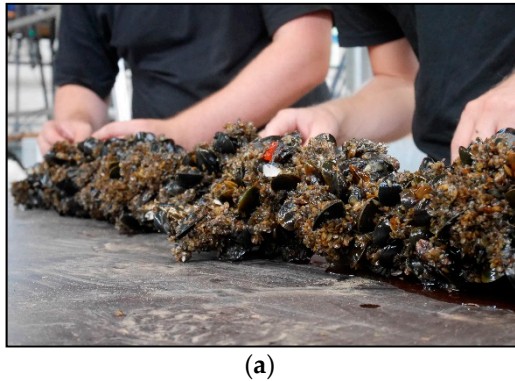
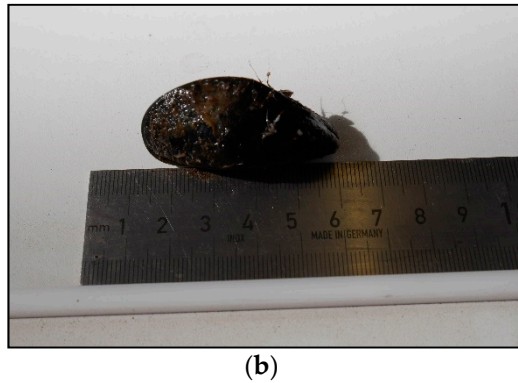

| (**a**) | (**b**) |

**Figure 4.** (**a**) A mussel specimen under preparation for drag and inertia testing and (**b**) an example of a single mussel randomly selected for data acquisition.

The specimens were inserted into the mounting clamps and tightened. For each specimen tested, several towing and wave tests with varying setups were conducted. However, before each set of experiments, an eigenfrequency test for the test rack itself was carried out with an impact hammer. The impact event was synchronously recorded by the 6-axis force transducer mounted to the load frame. By this means, the eigenfrequencies of the load frame as well as all possible combinations with dropper line specimens were recorded. Upon completion, the drag experiments in the wave tank were carried out with top- and bottom-mounted dropper lines that were towed at velocities of $u_1$ = 0.25 m/s, $u_2$ = 0.50 m/s, $u_3$ = 0.75 m/s and $u_4$ = 1.00 m/s. The mount at both ends ensures an equal flow velocity for the whole length of the dropper line. Each specimen, though differing in total length, was submerged over a length of $L_{wet}$ = 0.80 m. This, in turn, allows the correct determination of the drag coefficient $C_D$ for a specified length of mussel dropper line. The Reynolds numbers $Re = u_i * md/\nu$ covered during the tests were determined with the characteristic diameter $md$ = 10.31 cm, the aforementioned towing velocities $u_i$, the kinematic viscosity of the water $\nu$ = 1.004/998,200 m²/s and the constant water temperature of 20 °C and range from $2.0 \times 10^4$ to $1.1 \times 10^5$. In between all drag tests, the flow disturbances potentially induced by previous testing settled during a waiting period in order to avoid biased influence. Testing of the first specimen was repeated three times for testing repeatability. In order to gain further information about the force and motion response of the specimen, the towing operation was divided into forward and backward motions such that a single specimen is towed twice yet with a 180° change in direction. For specimens 2 and 3, the repetitions were reduced to one to allow for quicker laboratory tests, as the live mussels lose their cohesive properties when exposed to non-autochthonous conditions over a prolonged period of time [20]. After the completion of these tests, the setup of the dropper line was changed by loosening the lower clamp, leaving a top-mounted dropper line. As the pretests had revealed that major oscillations of the blue mussel dropper lines in the lateral and longitudinal directions occur at velocities $u > 0.50$ m/s, the tested velocities were reduced for the drag tests with the top-mounted specimens. The reduced velocities tested were $u_5$ = 0.10 m/s, $u_1$ = 0.25 m/s, $u_6$ = 0.375 m/s and $u_2$ = 0.5 m/s. Figure 5 shows the top-mounted mussel specimen dragged at $u_5$ = 0.10 m/s, $u_1$ = 0.25 m/s and $u_2$ = 0.5 m/s. The progressive lift towards the surface due to the forces acting on the dropper line is visible throughout the three sections.

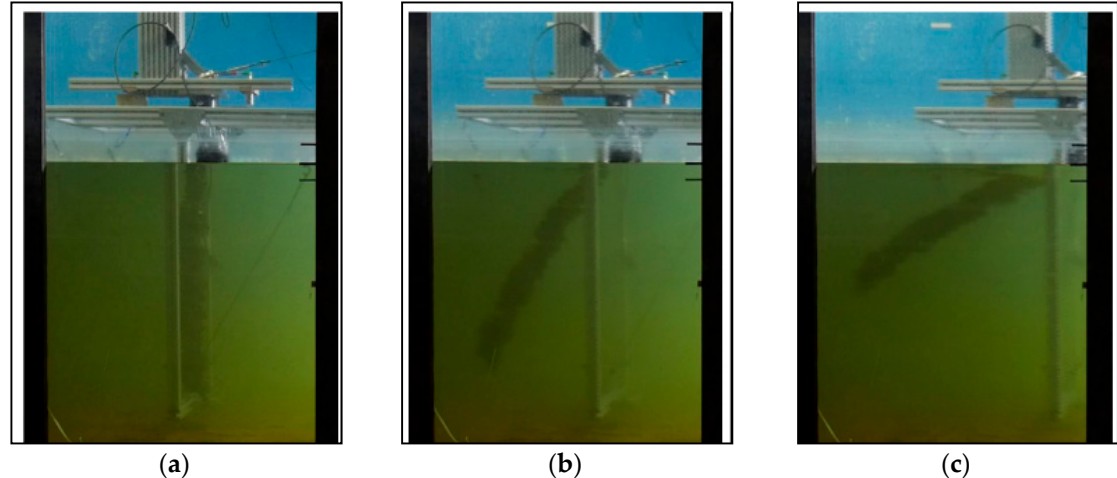

| (a) | (b) | (c) |

**Figure 5.** The drag testing of a top-mounted only specimen at velocities of 0.10 m/s (**a**), 0.25 m/s (**b**) and 0.5 m/s (**c**) with a progressive lift towards the surface visible.

After completing the towing tests, the carriage was positioned in front of a windowpane in the wall of the flume, which allows a side-view documentation of the motion response of the droppers during wave loading. The distance between the wave maker and carriage is 65 m. Three sets of waves were tested. These are described by their period $T$ and their targeted wave height $H$. Thus, the wave tests were carried out with targeted wave heights $H$ of 0.1 m, 0.12 m and 0.15 m with corresponding periods $T$ of 1.20 s, 2.4 s and 1.65 s. Each wave test was conducted once for each of the three specimens. The chosen values regarding the wave characteristics were related to potential offshore sites, e.g., off the coast of New Zealand or Canada, and scaled down to allow for future experiments with scaled surrogates. A 1:10 scale was selected, and a Froude similarity was applied. The corresponding Keulegan–Carpenter numbers $KC = u_i \times T/md$ ranged from 4.1 to 5.9. In Table 1, an overview of the wave height, wave period, maximum horizontal and vertical velocity, maximum horizontal and vertical acceleration and wave length is provided according to Stokes 2nd order wave theory for the waves tested in the flume. The tested wave heights are displayed in model $M$ and full-scale $F$ according to Froude similitude and a scaling factor of 1:10. The tests were performed for the top- and bottom-mounted configurations to enable comparative studies regarding the commonly investigated cylinders under wave loads as well as the free-swinging systems. Figure 6 exemplarily shows specimen 3 in waves with a targeted wave height of 0.12 m and a period of 2.4 s in a top-and-bottom-mounted configuration in comparison to a top-mounted only configuration. The latter setup results in an additional oscillatory movement of the blue mussel dropper line. To attain the actual forces acting on the blue mussel dropper line alone, frame-only tests are necessary as a concluding step. Here, all drag and inertia tests with corresponding current velocities and wave sets were carried out without an attached specimen.

**Table 1.** An overview of the wave height $H$, wave period $T$, wave length $L$, maximum horizontal $u_{max}$ and vertical $v_{max}$ velocity, and maximum horizontal $ax_{max}$ and vertical $az_{max}$ acceleration is provided according to Stokes 2nd order wave theory in model $M$ and full-scale $F$.

| Parameter $_{M/F}$ | Wave Test 1 | Wave Test 2 | Wave Test 3 |
|---|---|---|---|
| $H_M/H_F$ (m) | 0.10/1.00 | 0.12/1.20 | 0.15/1.50 |
| $T_M/T_F$ (s) | 1.20/3.80 | 2.40/7.59 | 1.65/5.21 |
| $L_M/L_F$ (m) | 2.23/22.3 | 6.63/66.3 | 3.92/39.2 |
| $u_{max,M}/u_{max,F}$ (m/s) | 0.26/0.82 | 0.21/0.66 | 0.31/0.98 |
| $v_{max,M}/v_{max,F}$ (m/s) | 0.08/0.25 | 0.13/0.41 | 0.29/0.91 |
| $ax_{max,M} = ax_{max,F}$ (m/s$^2$) | 0.44/0.44 | 0.45/0.45 | 1.18/1.18 |
| $az_{max,M} = az_{max,F}$ (m/s$^2$) | $-1.37/-1.37$ | $-0.41/-0.41$ | $-1.08/-1.08$ |

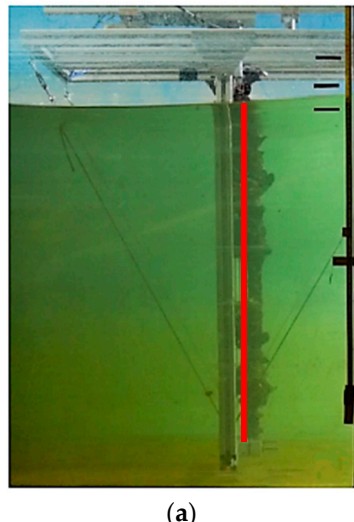 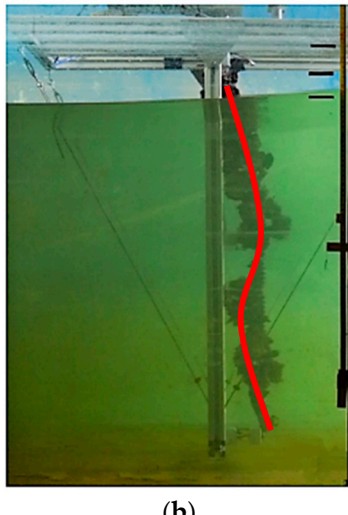

(**a**)            (**b**)

**Figure 6.** (**a**) The wave-loading on a blue mussel dropper line with a top-and-bottom-mounted clamp setup in contrast to (**b**) a top-mounted only clamp setup.

In between testing, a 3-D point cloud as a digital copy of the dropper line specimens was acquired by a terrestrial laser scanner in collaboration with the Geodetic Institute of the Leibniz Universität. The acquisition of the 3-D point cloud was carried out without immersing the specimen into the wave flume, as subaquatic scanning is less accurate and difficult to achieve with the current technology. The dropper lines were hung from the laboratory ceiling by a crane. The surrounding area is closed off for foot traffic to minimize specimen movement by unintended air flow. Around the blocked area, three tripods are pre-mounted to allow for an exact positioning of the 3-D laser scanner (Z+F IMAGER® 5010) with a measurement rate of up to 1 million points per second and a horizontal and vertical accuracy of 0.0007° (rms). Furthermore, reference points for the spatial registration were added in the vicinity of the scanning area. The 3-D point clouds from the differently positioned tripods were combined into a 3-D point cloud, with every single point containing information about x-, y-and z-directions as well as intensity.

## 3. Results and Discussion

### 3.1. Surrogate Creation

Mussels, as a structure, are subjected to natural fluctuations, e.g., crop coverage, diameter of the dropper line or mussel type. This is why the overall task of the model creation via the 3-D point cloud, acquired by the laser scanner, is to allow for the development of an artificial surrogate that features similar characteristics as compared to the live in situ specimen regarding the hydrodynamic behaviour in currents and waves. This means that the future testing of the surrogate models should lead to comparable drag and inertia coefficients $C_D$ and $C_M$ derived from the Morison equation [22]. The focus of this work is on the methodological aspects of the creation of the surrogates. Proceeding from the assumption that drag and inertia coefficients are significantly influenced by surface geometry, a precise description of the mussel surface is necessary for the design of a surrogate model. As a common surface descriptor, the Abbott–Firestone Curve (AFC) is selected for the 3-D point cloud of the mussel specimens as well as the surrogates. With this method, a thorough quantification of the surface geometry and porosity is possible [23]. In principle, the AFC displays the material distribution $M_r$ (%) as a function of the fluctuation in material surface $c$ (m). Mathematically, it is described by

$$M_r(c) = \frac{100\%}{l_n} * \sum_{i=1}^{n} l_i(c), \tag{1}$$

where $l_n$ is the total length of the recorded section and $l_i$ is the length of material cut at the depth $c$. The AFC separates its roughness profile into peak, medial and valley portions [23]. These separate planes differ in depth and main function, e.g., the peak portion has a major influence on a surface's run-in characteristics whereas the valley area defines the amount of water that may be dragged along when interacting with the flow. In case of relative movement between surfaces where friction is involved, the coefficient of friction changes over time as roughness peaks are diminished progressively. The depth of the peak, medial and valley portions is expressed by the parameters reduced peak roughness $R_{PK}$, medial roughness $R_K$ and reduced valley roughness $R_{VK}$ [24]. In order to achieve an averaging effect, unrepresentative peaks and valleys which make up less than two percent are disregarded [23].

The AFC enables the separate evaluation of a surface regarding run-in characteristics, the load-bearing capacity and the absorption of liquids. The parameters $M_{r1}$ and $M_{r2}$ describe the peaks' and valleys' surface portion in %. Originally designed as a descriptor for two-dimensional roughness profiles only, modern measurement techniques provide three-dimensional surface data, i.e., here, obtained from the laser scanning. Thus, a transfer of the Abbott–Firestone from 2-D to 3-D is necessary. Similar to the two-dimensional case, the 3-D AFC bases on an imaginary cutting-plane being steadily moved downwards from the profiles' highest peak to the lowest valley. This procedure is mathematically described by

$$SM_r(c) \;=\; \frac{100\%}{A} * \iint_{x,y} dx\,dy, \tag{2}$$

where $A$ is the regarded surface within the coordinate system $x$, $y$ [25]. The 2-D parameters reduced peak roughness $R_{PK}$, medial roughness $R_K$ and reduced valley roughness $R_{VK}$ may easily be transferred to 3-D by renaming them to the reduced peak roughness $S_{PK}$, the medial roughness $S_K$ and the reduced valley roughness $S_{VK}$ [26]. In 3-D, the material portions of peaks and valleys are named $SM_{r1}$ and $SM_{r2}$. After having recorded a three-dimensional surface's AFC, a comparison with others is possible. In the context of this work, the AFC is used as the primary surface descriptor and basis for all surrogate models. As this paper deals with cylinder roughness featuring a large unfiltered profile depth $P_t$ to a cylinder diameter $d$ ratio of

$$\frac{P_t}{d} \approx 0.5, \tag{3}$$

a definition of the AFC for cylindrical geometries is introduced. Instead of filtering out the cylinder's cylindricity and determining the material portion $SM_r$ as a function of height $c$ of a cutting plane, the cylindricity stays untouched allowing for the profile to be cut with a cylindrical surface $A_c(r)$, providing the material portion as a function of the cutting cylinder's radius $r$. Accordingly, the AFC for cylindrical bodies with a large unfiltered profile depth to cylinder diameter ratio may be defined as

$$SM_{r,cylindrical}(r) \;=\; 100\% * \frac{\sum A_i(r)}{A_c(r)}, \tag{4}$$

where $\sum A_i(r)$ is the profile's cut surface and $A_c(r)$ is the nominal surface of the cutting cylinder. This is depicted in Figure 7. Due to its promising characteristics regarding the surface description, the cylindrical AFC will be applied in the analysis of mussel crop surfaces.

After the acquisition of the 3-D point cloud with 5.4 million single points via the 3-D laser scanner, the data is treated via a statistical outlier removal filter in order to remove unwanted data points. Since the surface analysis via the cylindrical AFC assumes a solid body, the 3-D point cloud needs further processing. With regards to the limited computational resources, the 3-D point cloud is, therefore, divided into ten sections analyzed separately. To allow for further processing, all generated surfaces need to be closed and must not feature any holes. Due to the 3-D point cloud's limited quality resulting from challenging scan conditions (e.g., moving live mussels, wet and black surfaces to be scanned, shift

caused by moving air, etc.), the calculation of the desired surfaces requires compromises regarding shaping accuracy. The then solid bodies are analyzed with regard to the AFC. For the analysis, a total amount of 4,368,790 points within a total length of 0.915 m blue mussel dropper line is considered. The weighted arithmetic average material distribution is depicted in Figure 8. As can be seen in this figure, the mean specimen is 13.6 cm in maximum diameter and 3.3 cm in minimum diameter. Unrepresentative peaks and valleys which were not filtered by the outlier removal are not included in the results.

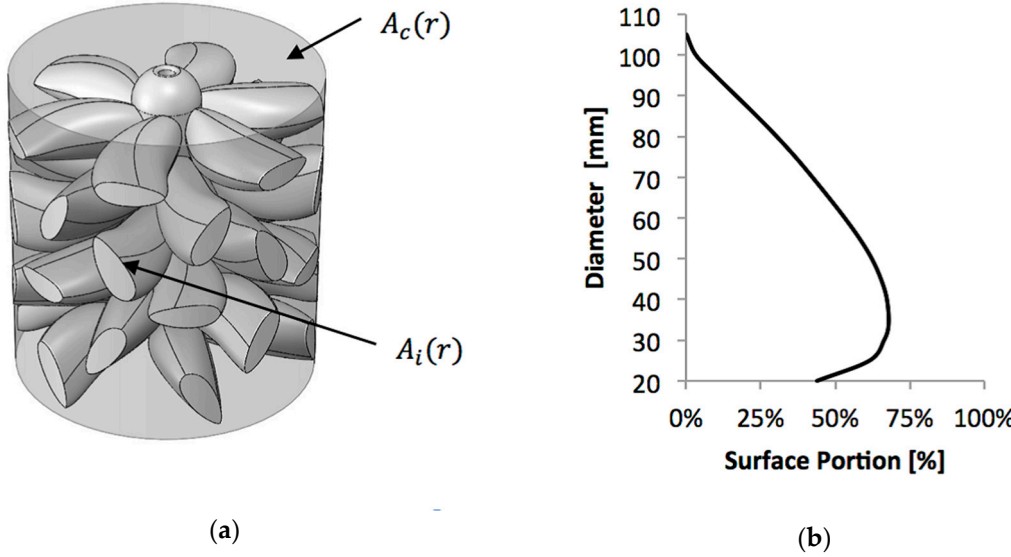

(**a**)                                                                (**b**)

**Figure 7.** (**a**) A visualisation for the creation of the Abbott–Firestone Curve for cylindrical surfaces: here are a mussel dropper line with $A_i(r)$, the profile's cut surface; $A_c(r)$, the nominal surface of the cutting cylinder; and (**b**) the resulting Abbott–Firestone Curve.

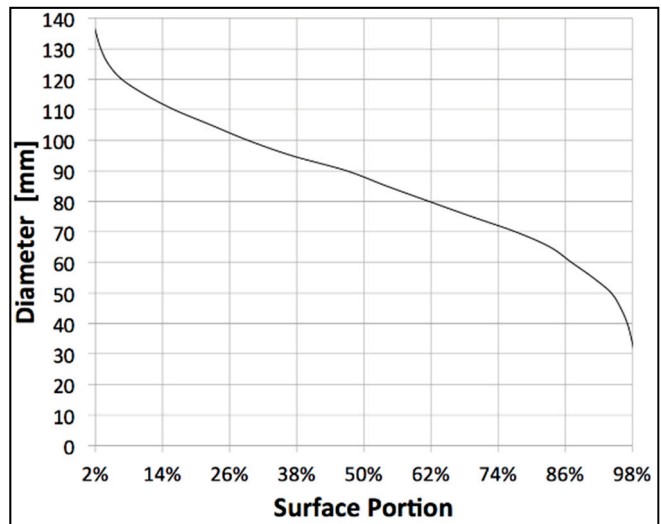

**Figure 8.** The weighted arithmetic average material distribution of all scanned sections.

The mean single mussel data mentioned before is the basis for the first concept of surrogate models. A uniform mussel is designed that equals the mean mussel's volume, weight, length and width. With the addition of the live specimens' statistical mean diameter of 10.32 cm for the dropper line sections, a design is possible. Copies of the uniform mussel are added to a slender cylinder at different angles of incidence until the mean weight per unit length is equal to the original live mussel data recorded. The angles of incidence are described as 0°, 90° and alternated-60°, whereby 0° is represented by mussels oriented horizontally, 90° by mussels oriented vertically and alternated-60°

by mussels changing their orientation by 60° in regard to the mussel above and below. The material distribution of the concept in comparison to the weighted arithmetic average material distribution of all sections is shown in Figure 9a alongside a side view of the alternated-60° surrogate.

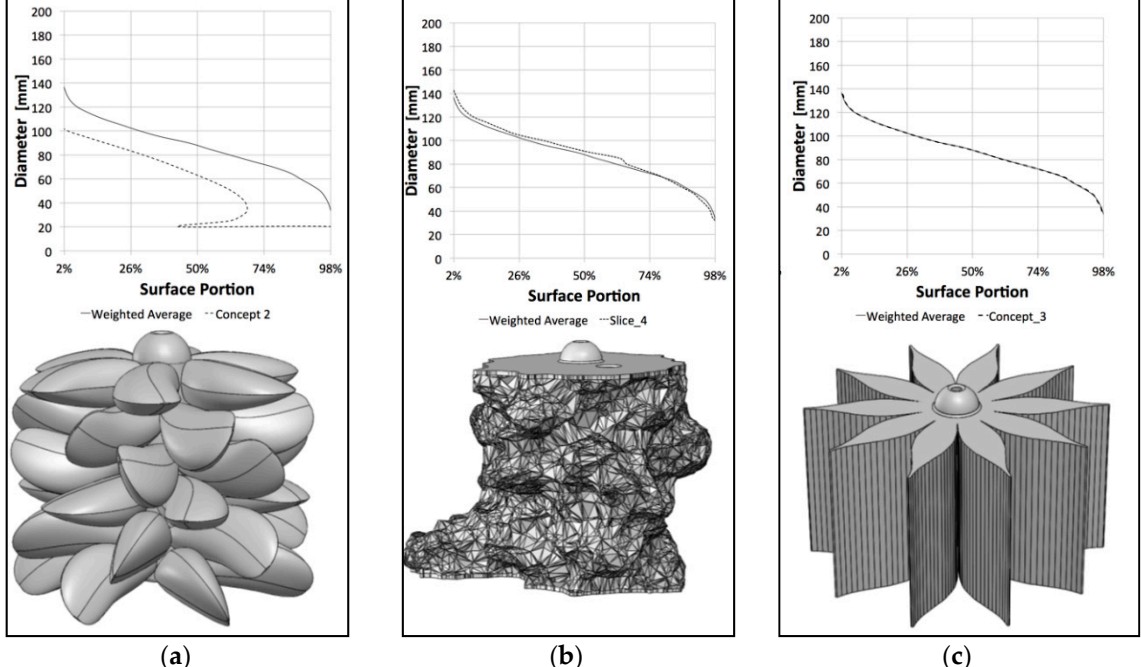

**Figure 9.** (**a**) Concept 1, (**b**) Concept 2 and (**c**) Concept 3 material distributions compared to the weighted arithmetic average material distribution of all sections.

The second concept for a surrogate model is based on the closest fitting 3-D point cloud section to the weighted arithmetic average material distribution. This is selected because its mass distribution features the most similarity to the original. A comparison of both AFCs as well as an exemplary side view of the surrogate model can be seen in Figure 9b.

The last concept is a reproduction of the weighted arithmetic average material distribution through a much simpler geometry that allows for easier scaling while maintaining the same AFC. The material distribution is closely fitted to the original, while the resemblance to the original live mussels is not obvious. Figure 9c, again, depicts the comparison between the two AFCs and a side view of the artificial surrogate model based on the direct AFC fit.

The resulting curves of Figure 9 display that the actual material distribution as a function of the fluctuation in material surface c (m) can be evaluated and that an evaluation by subdividing the profile into peak, medial and valley portions is possible. The AFC is applicable in 2-D and 3-D and additionally provides information on material-free and material-filled volumes. Even though the AFC is appropriate for a nonambiguous characterization of an existing surface, the reproduction of a certain surface based solely on the AFC is ambiguous. This can be seen in a comparison of the second and third surrogate concepts. The second, 3-D point cloud concept represents a close fit to the AFC, while the third concept represents the most exact results and a direct fit to the AFC. Thus, similar curves can be used to generate a variety of surrogate structures. For this reason, additional characteristics of a surface that relate to a certain Abbott–Firestone Curves should be considered. This could be the amount of break-off points along the structure or the solidity of the profile in flow. Eventually, energy dissipation and wake generation along the structure need to be better understood to find adequate additional surface descriptors. As a first step though, the above described surrogates are considered a promising starting point towards the creation of a mussel equivalent model that can be used for research regarding the behavior of suspended long-lines in current and wave conditions. In a last step, the different surrogate models are manufactured via selective laser sintering, an additive

manufacturing technique. This combines a cost-effective as well as rapid prototyping possibility for the production of the surrogates. The surrogates with a paint finish for better distinguishability are depicted in Figure 10.

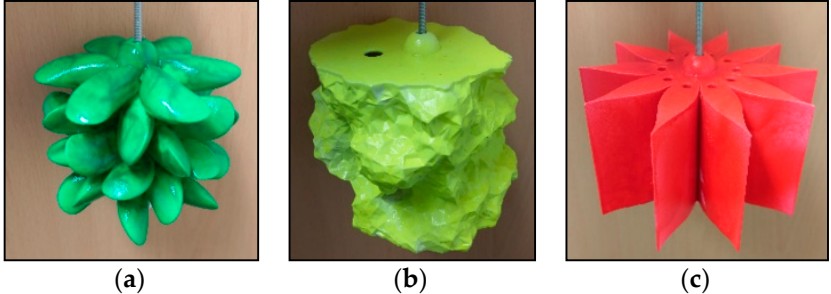

| (a) | (b) | (c) |

**Figure 10.** (**a**) Concept 1 with alternating-60° angle of incidence, (**b**) concept 2 with fit to weighted arithmetic average material distribution and (**c**) Concept 3, the direct Abbott–Firestone fit.

*3.2. Drag and Inertia Coefficients*

For a comparison of the created surrogates with the live specimen and an evaluation regarding the goodness-of-fit, an analysis of the hydrodynamic coefficients is needed in the future. The main focus of this work was the creation of the mussel surrogates; however, preliminary results for the live mussels were made available. The drag and inertia coefficients $C_D$ and $C_M$ for the live mussels under wave and current forcing were determined according to the Morison equation [22]. To this end, the time series of the forces in x-, y- and z-directions and the corresponding velocity of the towing carriage $u_i$ and profiling velocimeter were substituted into the Morison equation to determine the drag and inertia coefficients [22]:

$$F = \frac{1}{2}\rho C_D u^2 A + \frac{\pi}{4}\rho C_M V\dot{u} \tag{5}$$

where $F$ is the total horizontal force acting on the front face of a structure; $\rho$ = 1000 kg/m$^3$ is the density of fresh water; $C_D$ is the drag coefficient; $u$ is the horizontal particle velocity; $A = L_{wet} * D_i$ is the projected area with reference to the flow composed of the wetted length of the tested structure $L_{wet}$ and the characteristic, or mean, diameter $D_i$; $C_M$ is the inertia coefficient; $V$ is the volume of the structure; and $\dot{u}$ depicts the flow particle acceleration. The first term of Equation (5) corresponds to the drag forces, while the second term describes the inertia forces attributed to the flow acceleration. These terms can be isolated and the drag and inertia coefficients be determined via

$$C_D = \frac{2F}{\rho A u^2} \tag{6}$$

$$C_M = \frac{4F}{\pi\rho A^2 \dot{u}} \tag{7}$$

The tests are conducted with fresh water which has a negligible impact on the measurements as the values for CD and CM are independent of the density of the water. Tests with the same parameters, methodology and the surrogate models will be made available in upcoming research to determine the performance of the surrogates compared to the live mussels.

The preliminary results for the mean drag and inertia coefficients $C_D$ and $C_M$ with standard deviation and corresponding *KC* number as well as Reynolds number *Re* are shown in Table 2; Table 3. For the live mussel dropper lines tested in between Reynolds numbers of 2.0 × 104 to 1.1 × 105, resulting drag coefficients of $C_D$ = 1.16–3.03 were recorded. It can be seen that the coefficients for smaller Re-numbers are increasing. In earlier works and the absence of experimentally derived values, the drag coefficients of mussel dropper lines were assumed to be similar to ultra-rough cylinders where $C_D$-values of 1.7 and lower were put forward [11,12]. A more precise study into the drag

characteristics of mussel dropper lines was carries out by Plew et al. (2009) [13]. In the tests by Plew et al., a rigid, artificial mussel crop rope constructed from the shells of *Perna canaliculus* was used and a drag coefficient of 1.3 was determined within a Re-number range of 10,000 to 70,000. These values were used in a study by Dewhurst (2016) to determine the loading on a mussel raft [8]. It was concluded that these drag values may be too low for smaller Re-numbers. This is perpetuated by the results of this work where higher drag coefficients are recorded for the tests at lower velocities. The scope of this study focuses on the creation of a mussel surrogate; this is why further results and conclusions drawn regarding the $C_D$-coefficients and the drag tests can be seen in a work by Hildebrandt et al. [23]. The obtained CM-coefficients are still part of ongoing research, and the high standard deviation indicates that further investigations are necessary to accurately determine the inertia coefficient under waves. To the authors' knowledge, no comparable data for the inertia coefficient of mussel dropper lines is available as of now.

**Table 2.** The preliminary results regarding the mean $C_M$-coefficients with standard deviations with Re-numbers for drag tests.

| $Re= \frac{u*d}{v}$ | $\text{mean}C_D \pm SD$ |
|---|---|
| $2.0 \times 10^4$–$3.5 \times 10^4$ | $2.18 \pm 0.65$ |
| $3.5 \times 10^4$–$6.2 \times 10^4$ | $1.68 \pm 0.23$ |
| $6.2 \times 10^4$–$8.6 \times 10^4$ | $1.60 \pm 0.24$ |
| $8.6 \times 10^4$–$1.1 \times 10^5$ | $1.46 \pm 0.23$ |

**Table 3.** The preliminary results regarding the mean $C_M$ ranges with standard deviations with $KC$—numbers for wave tests.

| $KC= \frac{V*T}{L}$ | $\text{mean}C_M \pm SD$ |
|---|---|
| 4.10 | $1.05 \pm 0.81$ |
| 5.50 | $1.68Y \pm 0.89$ |
| 5.90 | $1.41 \pm 1.08$ |

## 4. Conclusions

This work aims at a better design and the facilitation of more efficient testing for new aquaculture concepts for usage in offshore environments. The overarching aim of the live-mussel testing is the identification of drag, inertia and turbulence characteristics of suspended mussel dropper lines. To this end, current and wave tests were conducted with three specimens of blue mussels. Drag testing was carried out at four different velocities of $u_1 = 0.25$ m/s, $u_2 = 0.50$ m/s, $u_3 = 0.75$ m/s and $u_4 = 1.00$ m/s, and the wave tests regarding the inertia characteristics were conducted with three different wave setups with targeted wave heights $H_1 = 0.10$ m, $H_2 = 0.12$ m and $H_3 = 0.15$ m and corresponding periods $T_1 = 1.20$ s, $T_2 = 2.40$ s and $T_3 = 1.65$ s. From this data, the following results regarding $C_D$- and $C_M$- coefficients were attained:

- First, results regarding drag coefficients showed that the mussel dropper lines tested in between Reynolds numbers of $2.0 \times 10^4$ to $1.1 \times 10^5$ showed $C_D$-coefficients of 1.16–3.03.
- The results regarding inertia coefficients showed that the mussel dropper lines tested in between KC-numbers of 4.1 to 5.9 showed $C_M$-coefficients of 0.25–1.25.

Furthermore, a 3-D laser scan of the mussels was conducted, which resulted in the generation of a 3-D point cloud of 5.4 million data points. A systematic approach employing the Abbott–Firestone Curve was developed that allows an analysis regarding the material distribution as a function of the fluctuation in the material surface. With this approach, three concepts for surrogate models were developed and will be subsequently tested:

- Concept 1 was based on single, uniform mussels. These were added to a slender cylinder at different angles of incidence until the mean weight per unit length was equal to the original live mussel data recorded.
- Concept 2 was based on the closest fitting 3-D point cloud section of the live mussels in regard to the weighted arithmetic average material distribution.
- Concept 3 was a reproduction of the weighted arithmetic average material distribution through a much simpler geometry.

The developed surrogates featured similar characteristics to the live mussels in regard to the chosen surface descriptor. In the future, a validation of the hydrodynamic characteristics is necessary to provide a scalable surrogate dropper line that can be used in a variety of applications. The three created surrogates will allow for further testing without constraints regarding the longevity of the mussels and will be evaluated regarding their comparability to the live-mussel data. The use of a large-scale flume facility to obtain data regarding full-scale offshore conditions is worth investigating. Tests of the single surrogates under current influence as well as full-scale testing under current and wave influence have been carried out. During the resulting evaluation, a best-fit surrogate will be created and used for further testing. This allows for a better understanding of the forces acting on suspended mussel dropper lines. The results will help optimization existing and emerging aquaculture systems by finding the best orientation of the farm layout to the prevailing hydrographic conditions while also finding modern, innovative system designs adapted to high energy environments.

**Author Contributions:** Conceptualization, N.G. and A.H.; Data curation, J.L., T.O. and J.-A.P.; formal analysis, J.L. and T.O.; funding acquisition, N.G. and A.H.; investigation, J.L. and T.O.; methodology, N.G. and A.H.; project administration, N.G. and A.H.; supervision, N.G. and A.H.; visualization, J.L. and T.O.; writing—original draft, J.L. and T.O.; writing—review and editing, J.L., T.O., N.G., K.H., B.H.B., J.-A.P. and A.H.

**Funding:** This Research has been supported with funding from the New Zealand Ministry of Business, Innovation and Employment through Cawthron Institute project CAWX1607.

**Acknowledgments:** The authors gratefully acknowledge the support of Tim Staufenberger for providing us with the mussel specimens.

**Conflicts of Interest:** The authors declare no conflict of interest.

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
