# Peer review of "Physical Modelling of Blue Mussel Dropper Lines for the Development of Surrogates and Hydrodynamic Coefficients"

_jmse, doi:10.3390/jmse7030065_

Round 1

Reviewer 1 Report

The manuscript reports on laboratory work on musell line hydrodynamics as well as technical work designed to provide 3D models of line structures.  The issue of drag in unsteady flows is important a certainly worthy of study.

I guess I don't get what is going on in this manuscript - it contains two parts - the Cd & Cm work and the scanning and surface analysis work.  I don't think they should be in the same paper - or if they are - the scanning work should be to provide proxies that are then compared with the real samples.

Then in line 263 -when we get to the list of what the drag coefficients are - we then are referred to another paper for final results and conclusions?  That doesn't make sense and needs to be reworked so as to make the present manuscript publishable.  What is the relevance of the present results and how do they help improve how we represent shellfish aquaculture structures in models?

Also I didn't get the bit about the wave characteristics gathered off the NZ coast (line 224-)... Putting aside the fact that the NZ work would have been in support of a different species of mussel you could come up with almost anything here.  It would be more useful to know about the range of conditions likely for this species and location... shellfish crop can find themselves in a wide range of conditions.

A GoPro camera gets used - these are impressive devices but do distort the image - was any correction applied?

Need to clarify that they are working on waves AND currents.

I was surprised to not see any reference to the work of David Plew - he did his thesis on aquaculture drag coefficients and a number of publications followed.

Plew, D.R., 2010. Depth-averaged drag coefficient for modeling flow through suspended canopies. Journal of Hydraulic Engineering, 137(2), pp.234-247.

Plew, D.R., Enright, M.P., Nokes, R.I. and Dumas, J.K., 2009. Effect of mussel bio-pumping on the drag on and flow around a mussel crop rope. Aquacultural engineering, 40(2), pp.55-61.

Plew, D.R., Stevens, C.L., Spigel, R.H. and Hartstein, N.D., 2005. Hydrodynamic implications of large offshore mussel farms. IEEE Journal of Oceanic Engineering, 30(1), pp.95-108.

Author Response

Dear Reviewer,

we would like to express our appreciation for the constructive comments raised by the editor, the other reviewers and yourself - we believe that by addressing them the quality of the revised manuscript has increased.

Please find enclosed below detailed answers to your comments, as well as, in the corresponding documnet, the actions performed to the originally submitted manuscript

Sincerely,

Jannis Landmann, Thorsten Ongsiek, Nils Goseberg, Kevin Heasman, Bela H. Buck, Jens-André Paffenholz, Arndt Hildebrandt

Reviewer 2 Report

Editing

LIne 18 - live bivalves (no hyphen)

Line 20 growing on dropper lines

Line 21 loads from currents and waves

Line 22 a custom made....was built to facilitate carriage-run gest with minimal delays

Line 35 protein

Line 42 near (no-)

Line 49 seafood "import trade" defecits

Line 42 no hyphen in in (-) and near (-)

Line 52 no hypen in (-), near (-)

Line 60 the mussels  (eliminate itself)

Line 104 - What is an interior profile?

Line 175 were they weighed in water also.

Comments

There are some published values of drag coeffiecients on mussel and should be included in the introduction and comparisons made in the discussion.

for semi-exposed areas (not inshore but no long period waves) you might consider the work of submersible mussel rafts

Wang, X., Swift, M. R., T. Dewhurst, I. Tsukrov, B. Celikkol and C. Newell. 2015. Dynamics of submersible mussel rafts in waves and current. China Ocean Engineering 29: 431:444.

Dewhurst, Tobias. "Dynamics of a submersible mussel raft." (2016). Ph. D. Thesis, University of New Hampshire.

Results - this paper would be greatly improved if it were possible to include the vertical velocities (z) measured in the tank since the mussel lines are stationary and the water is moving past the mussels as the waves go by.  this is because previous workers (Newell, C.R., D.J. Wildish and B.A. MacDonald.  2001. The effects of velocity and seston concentration on the exhalant siphon area, valve gape and filtration rate of the mussel Mytilus edulis. J. Exp. Mar. Biol. Ecol. 262: 91-111.) have show reduction in pumping rates when velocity is over 30 cm per second.  

Also, it would be worthwhile, if possible to list the acceleration of the water and the ropes related to the waves as this could also be important. The vertical acceleration of the ropes is related to potential drop-off.

The mussels are full sized but the waves are scaled down - nonetheless, it would be worthwhile to mention what each wave treatment is related to period in the open ocean. 

Perhaps a table showing : wave treatment  max vertical velocity max acceleration corresponding period in open ocean would be possible?

Author Response

(The authors gave the same response as above.)

Reviewer 3 Report

I found the text well written and straight to the point. Besides minor edits identified in the attached document,my main comment is that a couple of points should be included to the Discussion regarding the experimental conditions and how they relate to natural conditions. Details are also given in the attached document.

Author Response

(The authors gave the same response as above.)

Round 2

Reviewer 1 Report

The text is improved and you've satisfied most of my criticisms except the major one about the scanning and shape structure which apparently is going to be looked at elsewhere.

Either this should be removed (from line 316 on) or put it in front of the Cd work and connect to it when determining Cd.  At least give it a subheading.  As it is it seems like two completely disconnected pieces of work - the Cd material is useful but it is much less clear that the shape work is.  And if it is then this needs to be explained.

Line 247 typo "XY"

Author Response

(The authors gave the same response as above.)
